# Enhancing Methane Removal Efficiency of ZrMnFe Alloy by Partial Replacement of Fe with Co

**DOI:** 10.3390/molecules28114373

**Published:** 2023-05-26

**Authors:** Shumei Chen, Miao Du, Shuai Li, Zhinian Li, Lei Hao

**Affiliations:** 1GRINM Group Co., Ltd., National Engineering Research Center of Nonferrous Metals Materials and Products for New Energy, Beijing 100088, China; 13213556048@163.com (S.C.); shuail@grinm.com (S.L.); lzn@grinm.com (Z.L.); haolei@grinm.com (L.H.); 2GRIMAT Engineering Institute Co., Ltd., Beijing 101407, China; 3General Research Institute for Nonferrous Metals, Beijing 100088, China

**Keywords:** ZrMnFeCo, hydrogen purification, methane removal, XPS

## Abstract

High-purity hydrogen is extensively employed in chemical vapor deposition, and the existence of methane impurity significantly impacts the device performance. Therefore, it is necessary to purify hydrogen to remove methane. The ZrMnFe getter commonly used in the industry reacts with methane at a temperature as high as 700 °C, and the removal depth is not sufficient. To overcome these limitations, Co partially substitutes Fe in the ZrMnFe alloy. The alloy was prepared by suspension induction melting method, and was characterized by means of XRD, ICP, SEM and XPS. The concentration of methane at the outlet was detected by gas chromatography to characterize the hydrogen purification performance of the alloy. The removal effect of the alloy on methane in hydrogen increases first and then decreases with the increase in substitution amount, and increases with the increase in temperature. Specifically, the ZrMnFe_0.7_Co_0.3_ alloy reduces methane levels in hydrogen from 10 ppm to 0.215 ppm at 500 °C. ZrMnFe_0.7_Co_0.3_ alloy can remove 50 ppm of methane in helium to less than 0.01 ppm at 450 °C, demonstrating its excellent methane reactivity. Moreover, Co substitution reduces the formation energy barrier of ZrC, and Co in the electron-rich state demonstrates superior catalytic activity for methane decomposition.

## 1. Introduction

High-purity hydrogen has a multitude of applications in advanced optical materials for photodetectors and energy conversion. High-purity hydrogen is mainly used as carrier gas, reducing gas, coolant and insulating gas in the manufacture of optoelectronic devices. One significant application is the production of hydrogenated amorphous silicon (a-Si:H) films [1], widely used in photovoltaic cells and other optoelectronic devices. High-purity hydrogen serves as a carrier gas in chemical vapor deposition (CVD) processes, which deposit a-Si:H films with high purity and uniformity [2]. Additionally, it acts as a reducing gas in the synthesis of metal nanoparticles and other nanomaterials for energy conversion and photodetection applications [3]. Moreover, high-purity hydrogen acts as a coolant and insulating gas in advanced photodetectors such as superconducting single-photon detectors (SSPDs) and transition edge sensors (TESs) [4,5]. In these applications, even small amounts of impurities can have a detrimental effect on product quality or performance [6]. The presence of methane, one of the most common impurities in hydrogen gas, can lead to several issues. For instance, methane impurity can affect the morphology of graphene thin films, prepared as organic photovoltaic transparent electrodes by CVD using hydrogen as the carrier gas [7]. In addition, diamond prepared by CVD can be used in optical components for high-power and high-energy laser applications [8]. Hydrogen is accountable for the hydrophobic properties of the diamond surface and its p-type conductivity, and its purity also affects the optically active center [9]. Since semiconductor materials can be used in the preparation of lasers and detectors, methane causes defects in the semiconductor epitaxial diffusion process, affecting the quality of the final products [10]. Therefore, removing methane from hydrogen is critical in various applications.

Numerous purification technologies are employed to eliminate methane in hydrogen, including pressure swing adsorption [11], temperature swing adsorption [12], membrane separation [13], metal hydride [14], and getters [15]. PSA and TSA have become commonly used methods for large-scale crude hydrogen purification in the industry due to their advantages of occupying a large area, low cost, and ease of scale-up [16]. Through the selection of adsorbent and the control of operating parameters, hydrogen with a purity of 99.97% can be produced [17]. Palladium membrane separation, metal hydride, and getters can produce hydrogen with a purity higher than 99.9999% [18]. However, the cost of palladium membrane separation is high [19] and requires methane content in the raw gas to be less than 2 ppm [20]. Metal hydride method purifies through hydrogen adsorption and desorption, which has low purification efficiency and usually requires high temperatures to release hydrogen gas [18]. Among these, getters have distinct advantages in achieving deep purification of methane in hydrogen due to their high hydrogen purity, low cost, and portability. Getters are typically metals or alloys that use their high surface activity to physically adsorb gas molecules onto the surface of the alloy. Due to the affinity of the metal surface, molecules are dissociated into atomic states and dissolved into the surface layer of the metal, undergoing physical adsorption. Under the effect of the concentration gradient, they diffuse into the interior of the alloy, forming stable compounds [21]. One of the getters, SAES St909, is a ZrMnFe alloy getter material developed to dissociate tritiated water for tritium recovery in the Savannah River Site (SRS) tritium facility projects [22]. St909 is supplied as a single-phase ZrMnFe alloy with an aluminum binder for pellet formation, consisting of 40.5 wt% Zr, 24.5 wt% Mn, 25 wt% Fe, and 10 wt% Al [23]. It has shown high methane reactivity and can be used for gas purification. St909 (ZrM-nFe) getter was used to remove 0.1% and 1% of CH_4_ in helium. At 700 °C, the removal rate was 99%, while at 600 °C, the removal rates were 98% and 93%, respectively [24]. Methane decomposition efficiency ranged from 80% to 95% at 700 °C with nitrogen containing 2.5% methane. However, tests at 600 °C showed negligible methane removal, which is attributed to the competition of nitrogen and methane with the getter at high temperatures [25]. Bench-scale St909 tests provide relatively fast and inexpensive results on St909 methane cracking performance under different test conditions. On the other hand, pilot-scale St909 tests provide a better representation of St909 performance in a full-scale bed for a given set of test conditions. Full-scale beds contain nominally 5300 g of St909. In the pilot test, 20% hydrogen, impurity composition of 1.1% methane, and 0.4% nitrogen with a balance of helium, 30 sccm feed, 203 kPa (1520 torr) back-pressure, and 700 °C were employed. After purification, the outlet methane concentration can reach tens of ppm, demonstrating the excellent ability of the getter to decompose methane [26]. All tests were conducted at 700 °C at a pressure of 101.3 kPa (760 torr) with a 10 sccm feed of nominal 5 vol% methane (4.9 kPa partial pressure) in helium or hydrogen for 24 h. The optimal decomposition efficiency of the absorbent for methane in helium gas can reach 99%, whereas under the same conditions the decomposition efficiency for methane in hydrogen gas is only 79% [27]. Hydrogen affects the methane cracking efficiency of the getter by influencing the chemical equilibrium state of the methane decomposition reaction. Although ZrMnFe can purify hydrogen at high temperatures, it is highly unsafe to purify hydrogen in this way due to its flammability and explosiveness. Furthermore, it consumes a lot of energy to maintain high-temperature purification. Thus, enhancing the reactivity of the alloy with methane at low temperatures is crucial for the deep purification of hydrogen. Previous studies on the removal of ZrMnFe alloy mainly focused on high concentration methane, and there is a lack of research on removing low-concentration methane (ppm) in the hydrogen feed gas to the ppb level.

Element doping is a widely used technique for modifying alloys. Yu et al. addressed the issue of low methane decomposition rates in hydrogen gas at 600 °C for the ZrNi alloy by partially replacing its constituent elements. The resulting ZrTiNiMn alloy was tested for its adsorption performance against 1.01% and 0.2% methane in hydrogen gas at 650 °C, and it exhibited a methane removal efficiency of more than 94%. This reaction may be a mixed reaction involving the catalytic cracking of methane by Ni and ZrNi, as well as the direct reaction of some strong carbide-forming elements, such as V and Ti, in the alloy with methane [28]. Hence, one way to enhance the methane removal efficiency of the ZrMnFe alloy is by incorporating elements with superior catalytic performance for methane cracking. Co and Fe are both transition elements belonging to the same group, and the metal coordination bond dissociation energy formed between Co and methane ligand is relatively low, making it easier to activate methane and break the carbon–hydrogen bond [29]. Song et al. conducted a comparative analysis of the activation energies for the C-H bond breaking of methane on catalysts Ni, Co, and Cu. They discovered that the order of activation energy for methane decomposition was Ea(Co) < Ea(Ni) < Ea(Cu). The bond breaking rates followed the order k(Co) > k(Ni) > k(Cu), indicating that the catalytic performance of these transition metals can be ranked in order from high to low as Co > Ni > Cu [30]. To modify the ZrMnFe alloy, partial substitution of Fe with Co was studied. The substitution amount of Co ranged from 0 to 0.4 to avoid decreasing the hydrogen absorption platform pressure of the alloy, which may weaken its purification performance [31].

The purification of hydrogen has become a crucial area of research due to the growing demand for high-purity hydrogen in various industrial applications. The currently used ZrMnFe getter reacts with methane in hydrogen at a relatively high temperature, resulting in high energy consumption and potential safety hazards. Modified ZrMnFeCo alloys are a promising solution for this purpose, as they have the potential to eliminate impurities from hydrogen gas. However, their effectiveness in removing trace amounts of methane from hydrogen has not been extensively studied. The goal of this study is to investigate the efficiency and mechanism of removing trace amounts of methane from hydrogen using a modified ZrMnFe alloy, in which Fe partially replaces Co. Through this investigation, we aim to gain a deeper understanding of the factors that affect the performance of hydrogen purification technology and identify potential areas for improvement. This study is expected to contribute to the development of more efficient and effective hydrogen purification techniques.

## 2. Materials and Methods

ZrMnFe_1-x_Co_x_ (x = 0, 0.1, 0.2, 0.3, 0.4) alloys were prepared by partially replacing Fe with Co, which has higher catalytic activity for methane, in order to reduce the reaction temperature. The modified alloys were prepared by suspension induction melting method. The phase structure, composition, microscopic morphology and valence state of surface elements were characterized.

### 2.1. Material Preparation

The experiment used metal raw materials with a purity of over 99.5% which were purchased from Beijing Yanbang New Materials Technology Co., Ltd. The alloys were synthesized through the suspension induction melting method. The metal raw materials were placed in a water-cooled copper crucible in sequence, washed thrice with argon, evacuated to below 1 Pa each time, and then filled with argon to −0.05 MPa. Then, the alloy was induction melted under the protection of an argon atmosphere. After melting, the alloy ingots were cooled in the water-cooled copper crucible. The ingots were flipped and remelted three times to ensure uniformity in composition, and then annealed at 900 °C for 10 h to produce uniformly composed cast ingots. The alloys were crushed in an argon glove box and screened to obtain alloy particles ranging from 60 to 160 mesh. A total of 50 g of the alloy powder was placed in a self-made purification reactor and activated under vacuum at 450 °C for 2 h to remove surface passivation films, oxide films, and adsorbed gases. Once it had cooled to room temperature, the purified reactor was connected to the pipeline using standard gas through blowing. 

### 2.2. Material Characterization

The alloy material was characterized using several techniques. X-ray diffraction (XRD) was utilized to determine the phase structure of the alloy by comparing it with standard cards. The analysis was performed using an X’ Pert PRO MPD X-ray diffractometer with a scanning range of 20–80° and a scanning speed of 5°/min. Inductively coupled plasma emission spectrometry (ICP) was utilized to determine the content of metallic elements in the alloy. Scanning electron microscopy (SEM) was utilized to examine the microstructure of the alloy before and after testing at 200× and 2000× magnification, providing a detailed understanding of its morphology. X-ray photoelectron spectroscopy (XPS) was utilized to analyze the valence state of surface elements in the alloy, offering crucial information on its chemical composition.

### 2.3. Hydrogen Purification Performance Testing

The purification process was conducted in a laboratory-scale container with a welded seal at the lower end and a flange seal at the upper end. To prevent the sample powder from entering other equipment in the system, filter screens were inserted at both ends of the reaction vessel. The heating and insulation devices on the exterior of the reaction vessel were designed to ensure uniform heating and minimize heat loss during the purification process. The temperature control system was equipped with a high-precision temperature sensor and a feedback loop, which ensured accurate and stable temperature control. The testing equipment, as shown in Figure 1, included a gas supply system, a regulator, a purification reactor, a gas chromatograph equipped with a mass flow controller, and auxiliary devices such as valves, pipelines, and filters. A high-purity helium gas was used as both the carrier and driving gas in the DID detector-equipped gas chromatograph. This detector utilizes high-energy ultraviolet radiation (400–500 nm) to ionize impurity molecules in the sample, and then collects and amplifies their signals through a collector electrode, thereby obtaining the spectral peak of the measured component. During the test, the hydrogen standard gas containing 10 ppm methane was used to make the standard spectrum, and then the gas passed into the outlet of the reactor was integrated to obtain the response value so as to calculate the methane concentration. With a detection limit of 0.010 ppm, this system enabled the precise measurement of trace amounts of impurities in the hydrogen gas. Since it is dangerous to purify high-temperature hydrogen in a stainless steel reactor, we chose a reaction temperature of 300~500 °C and a reaction pressure of 0.1–0.5 MPa. The test results of the gas chromatograph used in the experiment are more accurate when the gas flow rate is between 70 and 100 sccm, so we chose the gas flow rate of 70 to 100 sccm for testing.

## 3. Results and Discussion

### 3.1. Annealed Alloy Characterization

To ensure uniformity of composition, the alloy ingots produced by suspension induction melting underwent uniform annealing to eliminate any precipitated phases. To prevent oxidation, the alloys were crushed within an argon glove box. The XRD pattern of the ZrMnFe_1-x_Co_x_ (x = 0, 0.1, 0.2, 0.3, 0.4) alloy sample is shown in Figure 2, confirming the formation of the C14 Laves phase with a P63/mmc space group, which is a typical structure for intermetallic compounds. The observed peaks in the XRD pattern matched well with the standard diffraction peaks of ZrMn_2_ (JCPDS #39-1031), indicating successful synthesis of the desired alloy composition with high structural homogeneity.

During the suspension induction melting process, Mn is easy to volatilize due to its high saturation vapor pressure, making it difficult to control its content. Therefore, an excess of 5% Mn was added to the raw materials [32], but the melting process parameters have some effect on the Mn content, resulting in an error. Table 1 displays the ICP results of the alloy samples, showing that the actual content of each element is consistent with the design value with an error of less than 5%.

Figure 3 shows the SEM images of the annealed alloys. The alloys exhibit irregular, block-shaped, and densely packed crystalline particles. The mechanical crushing process led to the dispersion of a significant number of small particles on the surface of the ZrMnFe_0.7_Co_0.3_ alloy. In comparison, the ZrMnFe alloy displayed better crystallinity, characterized by block-shaped particles with sizes in a range of several hundred micrometers. The introduction of Co caused an uneven surface morphology in the alloy, which could be attributed to the energy fluctuations that arose from the Co addition, consequently affecting the crystallinity of the alloy.

### 3.2. Hydrogen Purification Performance

The study evaluated the performance of hydrogen purification reactors containing ZrMnFe_1-x_Co_x_ (x = 0~0.4) alloys at different temperatures, a test pressure of 0.1 MPa, and a controlled gas flow rate of 70 sccm at the outlet. The methane concentration at the outlet was measured, and the results are presented in Figure 4. At 350 °C, the only purification columns that effectively removed methane were those containing ZrMnFe_0.7_Co_0.3_ and ZrMnFe_0.6_Co_0.4_, with a removal efficiency of approximately 36%. As the temperature increased above 400 °C, the purification effect gradually improved. Compared to the ZrMnFe purification reactor, the methane removal efficiency of the purification column with different Co substitution levels improved at the same temperature. Except for ZrMnFe_0.6_Co_0.4_, the methane removal efficiency of the purification reactor increased with increasing Co substitution level at the same temperature. At 400 °C, the purification column with a Co substitution level of 0.3 was able to reduce the concentration of 10 ppm CH_4_ in H_2_ to about 1.194 ppm, while the ZrMnFe alloy purification column only reduced the methane content to 7.146 ppm, which was not significantly different from the removal efficiency of ZrMnFe_0.9_Co_0.1_ and ZrMnFe_0.8_Co_0.2_. At 450 °C, the improvement in Co substitution on methane removal was significant. At this temperature, the adsorption efficiencies of ZrMnFe_0.9_Co_0.1_, ZrMnFe_0.8_Co_0.2_, ZrMnFe_0.7_Co_0.3_, and ZrMnFe_0.6_Co_0.4_ for 10 ppm CH_4_ in H_2_ were 67.15%, 86.52%, 93.80%, and 91.50%. The removal efficiency of all Co-substituted alloys is higher than that of ZrMnFe. The removal efficiency of all Co-substituted alloys is higher than that of ZrMnFe. At 500 °C, the adsorption efficiency further increased to 83.77%, 91.12%, 97.08%, and 93.70%, respectively. The methane adsorption efficiency of the modified alloy increased with the increasing temperature.

The actual methane content in the hydrogen standard gas was 9.56 ppm, and the standard curve was established after three gas chromatography tests. The gas chromatogram of the standard gas is presented in Figure 5, where the methane peak is observed at approximately 11.8 min, with a response signal of 290.359 eV. The methane concentration at the outlet of the ZrMnFe_0.7_Co_0.3_ reactor was measured to be 0.292 ppm, with a peak response signal of 8.875 eV, under the conditions of 0.1 MPa pressure, 70 sccm gas flow rate, and 500 °C temperature.

Figure 6a reveals an inverse correlation between hydrogen purification efficiency of the ZrMnFe alloy and inlet pressure ranging from 0.1 to 0.5 MPa. This phenomenon is in line with Le Chatelier’s principle, wherein the increase in pressure drives the equilibrium of the methane decomposition reaction to shift in the reverse direction. In contrast, the ZrMnFe_0.7_Co_0.3_ alloy exhibits a distinct trend wherein its purification efficiency is directly proportional to the inlet pressure. This slight enhancement in efficiency can be attributed to the increased catalytic activity on the alloy’s surface resulting from the Co substitution, leading to an upsurge in the methane decomposition rate. Furthermore, a rise in pressure intensifies the decomposition reaction and improves its effect by causing more frequent collisions between methane molecules and the alloy surface. At this juncture, the ZrMnFe_0.7_Co_0.3_ alloy can absorb 10 ppm of methane in hydrogen to 0.215 ppm. In Figure 6b, an increase in gas flow rate results in an increased concentration of outlet methane and a corresponding decrease in the purification effectiveness of the alloy. The reduction in methane removal effect is due to the shorter contact time between the gas and the alloy at higher flow rates, which hinders the complete adsorption of methane. 

The pressure and temperature limits of the flange-sealed reactor used in the experiment have not been thoroughly investigated. Due to safety concerns in testing high-temperature hydrogen gas, the testing pressure used is relatively low. Due to the limitation of the gas chromatograph on the gas flow rate, the gas flow rate ranges from 70 to 100 sccm. In practical applications, in order to ensure the supply of high-purity gases, it is necessary to monitor the purity of the gas, and the purification reactor should be replaced in time if the purity is lower than expected. When assembling the purification unit at the end, the cleanliness of the pipeline and valves should be ensured to avoid introducing impurities. In addition, due to the high surface activity of the adsorbent material, both end valves should be closed in time when not in use to avoid oxidation and failure of the material exposed to the air.

### 3.3. Co Element Catalytic Performance Verification

To investigate the catalytic effects of Co and to eliminate hydrogen interference on the chemical equilibrium, a pure-gas purification test was conducted using a standard gas mixture of 50 ppm CH_4_ in helium. Figure 7 shows the evaluation of CH_4_ adsorption performance of the ZrMnFe and ZrMnFe_0.7_Co_0.3_ alloys at different temperatures, as shown in Figure 7. At a lower temperature of 350 °C, the substitution of Fe with Co did not significantly improve the catalytic performance. However, at 400 °C, the Co-substituted alloy exhibited enhanced CH_4_ catalytic cracking performance, adsorbing 50 ppm CH_4_ in helium gas to around 0.1 ppm. At 450 °C, the ZrMnFe_0.7_Co_0.3_ purification reactor adsorbed 50 ppm CH_4_ in helium gas to below the chromatographic detection limit (less than 0.01 ppm), demonstrating superior catalytic activity compared to the ZrMnFe purification reactor that achieved CH_4_ adsorption of only about 0.4 ppm at that temperature. The results indicate that the Co substitution in the ZrMnFe_0.7_Co_0.3_ alloy significantly enhances its catalytic performance for CH_4_ adsorption and cracking, leading to more effective purification of the gas mixture.

### 3.4. Reaction Mechanism

After testing the hydrogen purification process containing 10 ppm CH_4_ at 500 °C, it was observed that the ZrMnFe alloy maintained its ZrMn_2_ structure, with the XRD analysis indicating the presence of elemental carbon, as shown in Figure 8. In contrast, the XRD results for the ZrMnFe_0.7_Co_0.3_ alloy exhibited the characteristic peak of ZrC, implying that the alloy formed a metal carbide with the decomposed products during purification. Notably, the absence of elemental carbon in the XRD pattern suggests that the ZrMnFe_0.7_Co_0.3_ alloy has a faster rate of ZrC formation after methane decomposition compared to the ZrMnFe alloy. This behavior implies that the reaction between the ZrMnFe_0.7_Co_0.3_ alloy and methane is primarily governed by the rate of ZrC formation, whereas the reaction between the ZrMnFe alloy and methane is regulated by the surface methane decomposition rate. The partial substitution of Co for Fe is likely to be the reason for this phenomenon since it affects the electronic structure of Zr and reduces the formation energy barrier of ZrC. Therefore, the ZrMnFe_0.7_Co_0.3_ alloy is still in the phase of rapid methane adsorption.

After evaluating the performance of the hydrogen purification process, it was observed that the particle size of the alloy decreased, as shown in Figure 9. Interestingly, the pulverization phenomenon of the ZrMnFe_0.7_Co_0.3_ alloy was more severe than that of ZrMnFe under similar conditions. A high magnification scanning electron microscope revealed numerous cracks in the ZrMnFe_0.7_Co_0.3_ particles. The cracks exposed more active surfaces, which increased the contact area between the reaction gas and the alloy surface. This phenomenon led to an improvement in the reaction efficiency of the purification process. The increased surface area facilitated better interaction between the reaction gas and the alloy, allowing for the removal of impurities from the hydrogen.

We conducted X-ray photoelectron spectroscopy (XPS) to characterize the surface elements Zr, Fe, and Co of the annealed and tested ZrMnFe and ZrMnFe_0.7_Co_0.3_ alloys. Our analysis of the peak shifts of characteristic peaks revealed the effect of Co substitution on the purification performance of the alloy. As shown in Figure 10, the peaks near 181.5 eV and 184.5 eV correspond to ZrO_2_, which may result from the surface Zr element being oxidized during sample testing. After Co substitution, the characteristic peak position of Zr moved to a lower binding energy, possibly due to an increase in valence electrons. This lead to better shielding effects and reduced the electron binding energy, making Zr more prone to react with carbon and form metal carbides. Therefore, a characteristic peak of ZrC appeared at 179.57 eV after reaction in ZrMnFe_0.7_Co_0.3_. The Fe 2p orbital spectrum peak showed two Fe^3+^ peaks at the binding energies of 712.7 eV and 725.0 eV, respectively, due to the oxidation of Fe during XPS testing. After Co substitution, the characteristic peak shifted to a lower binding energy, indicating an increase in the electron cloud density around Fe originating from the transfer of electrons in Co. After the reaction, the characteristic peak of Fe in ZrMnFe shifted to a lower binding energy, while the characteristic peak of Fe in ZrMnFe_0.7_Co_0.3_ shifted to a higher binding energy, indicating that Fe lost electrons in ZrMnFe_0.7_Co_0.3_. This made the carbon in methane more likely to receive electrons and become elemental carbon. The Co 2p orbital characteristic peak corresponds to the Co 2p_3/2_ and Co 2p_1/2_ orbitals, and the positions of the satellite peaks at 780.11 eV and 795.85 eV correspond to metallic Co. After the purification performance test, the binding energy shifted to the low-field direction, indicating an increase in external electron density. This electron-rich state of Co is more conducive to activating the C-H bond in methane.

## 4. Conclusions

The preparation of the ZrMnFeCo alloys was conducted using a suspension induction melting method to produce alloys with varying levels of Co substitution, ranging from 0 to 0.4. The resulting modified alloy had a phase structure and elemental content matching the expected design values, indicating the effectiveness of the synthesis method. Moreover, the modified alloy demonstrated improved performance in purifying hydrogen gas, specifically in removing trace amounts of methane. At 500 °C, the ZrMnFe purification reactor reduced methane levels from 10 ppm to about 4 ppm, whereas the ZrMnFe_0.7_Co_0.3_ alloy purification reactor significantly reduced methane levels from 10 ppm to 0.215 ppm, demonstrating its practical effectiveness. The purification efficiency of the alloy increases with increasing temperature, and the purification efficiency of ZrMnFeCo is higher than that of ZrMnFe at the same temperature. The methane purification efficiency of the ZrMnFe alloy decreases with increasing pressure, while the methane purification efficiency of the ZrMnFe_0.7_Co_0.3_ alloy slightly increases with increasing pressure. An increase in gas flow rate results in an increased concentration of outlet methane and a corresponding decrease in the purification effectiveness of the alloy. In the purification performance of helium containing 50 ppm methane impurity, the enhanced ability of the alloy to decompose CH_4_ at low temperature after Co substitution was reflected. The observed enhancement in purification efficiency can be attributed to the partial substitution of Co for Fe, which resulted in a reduction in the size of the alloy particles, thereby increasing the number of active surfaces. Additionally, the Co substitution was found to lower the formation energy barrier of ZrC. Furthermore, Co in the electron-rich state exhibited superior catalytic activity for methane decomposition, highlighting the potential of these modified alloys for further development in hydrogen purification technology.

Modified alloys can produce hydrogen gas with a purity higher than 6 N at 500 °C, which can be directly used in the relevant application fields. Meanwhile, after purification, the methane impurity concentration is lower than 2 ppm, making it possible to combine with the palladium membrane separation method to produce hydrogen gas with even higher purity and reduce the poisoning effect of methane impurities on the palladium membrane. Additionally, further research is needed to investigate the impurity adsorption capacity of the getter, and the effect of other impurities on the purification performance is also a research focus for the practical application of the alloy.

## Figures and Tables

**Figure 1 molecules-28-04373-f001:**
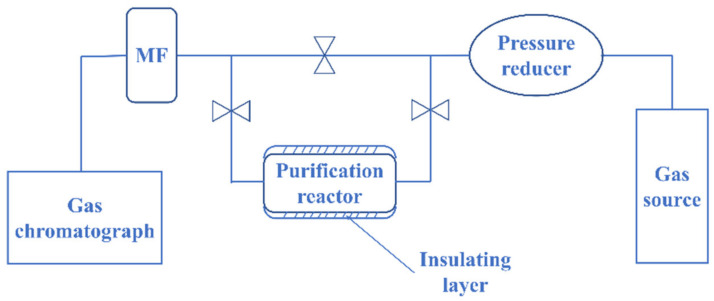
Purification reactor test device diagram.

**Figure 2 molecules-28-04373-f002:**
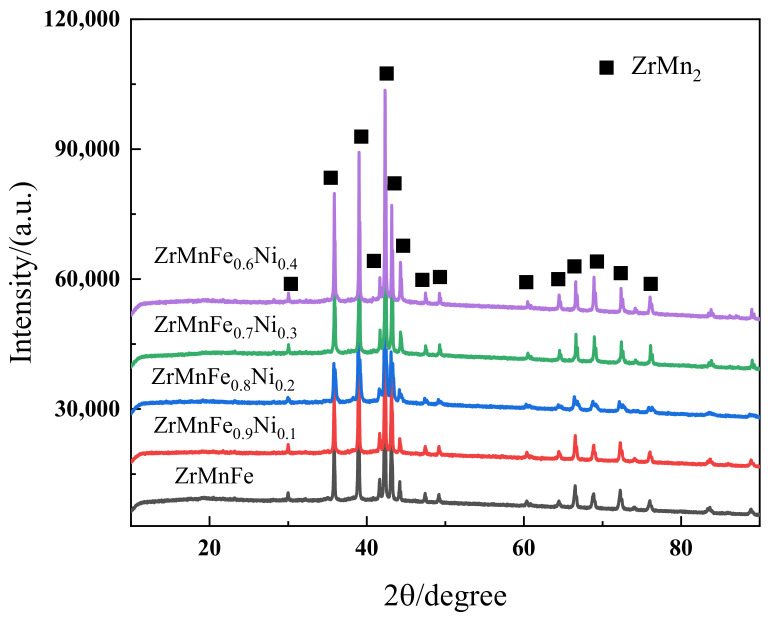
XRD pattern of the ZrMnFe_1-x_Co_x_ (x = 0, 0.1, 0.2, 0.3, 0.4) alloy.

**Figure 3 molecules-28-04373-f003:**
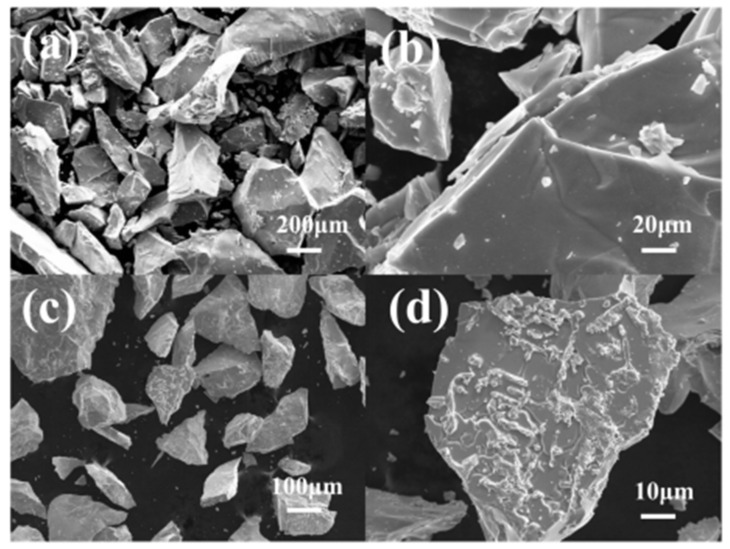
SEM images of alloys: (**a**,**b**) Annealed ZrMnFe alloy; (**c**,**d**) Annealed ZrMnFe_0.7_Co_0.3_ alloy.

**Figure 4 molecules-28-04373-f004:**
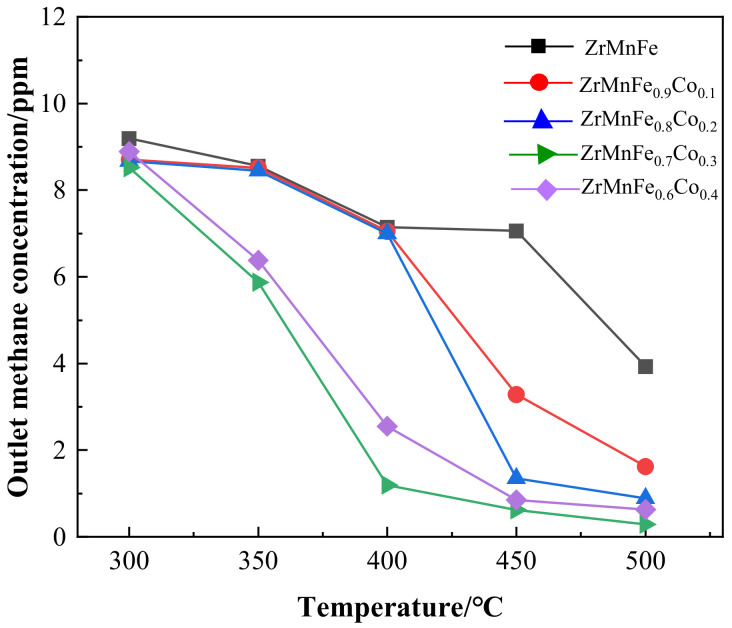
The relationship between methane concentration and temperature for the ZrMnFe_1-x_Co_x_ (x = 0~0.4) alloys.

**Figure 5 molecules-28-04373-f005:**
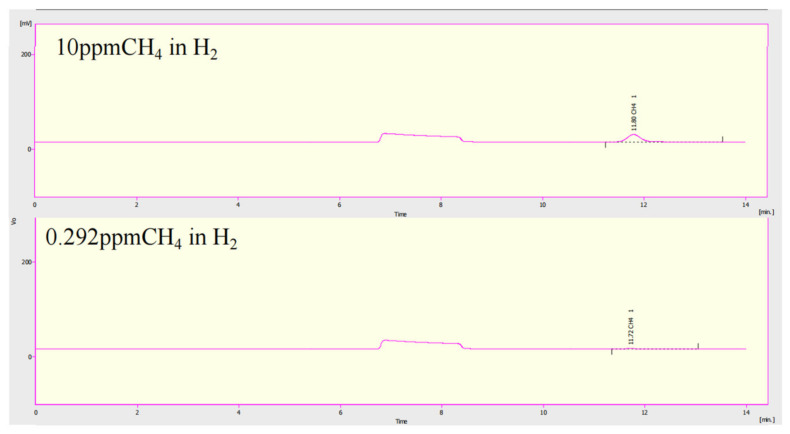
Gas chromatogram.

**Figure 6 molecules-28-04373-f006:**
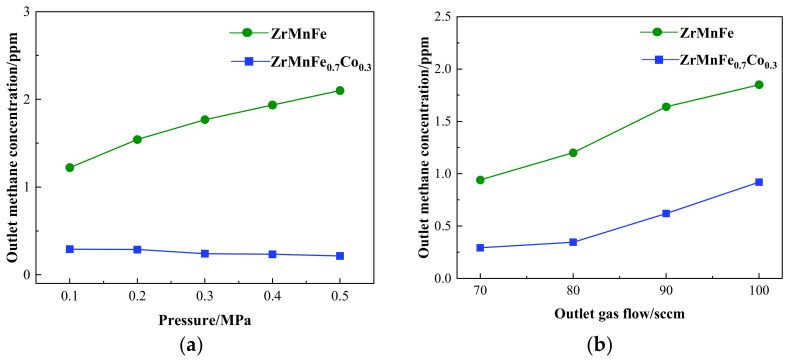
(**a**) The relationship between methane concentration and pressure for ZrMnFe and ZrMnFe_0.7_Co_0.3_ at 500 °C and 70 sccm; (**b**) The relationship between methane concentration and gas flow for ZrMnFe and ZrMnFe_0.7_Co_0.3_ at 500 °C and 0.1 MPa.

**Figure 7 molecules-28-04373-f007:**
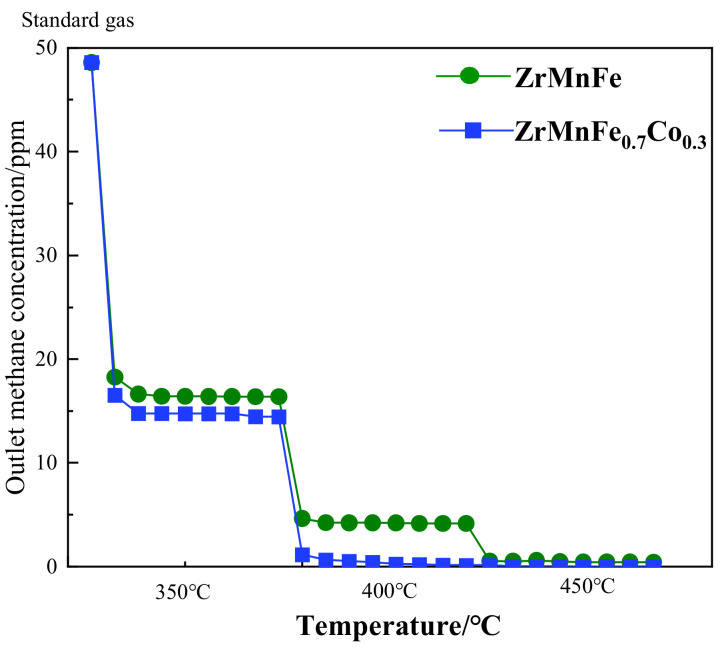
The curve of outlet methane concentration–temperature–sampling time for the ZrMnFe and ZrMnFe_0.7_Co_0.3_ alloy in He.

**Figure 8 molecules-28-04373-f008:**
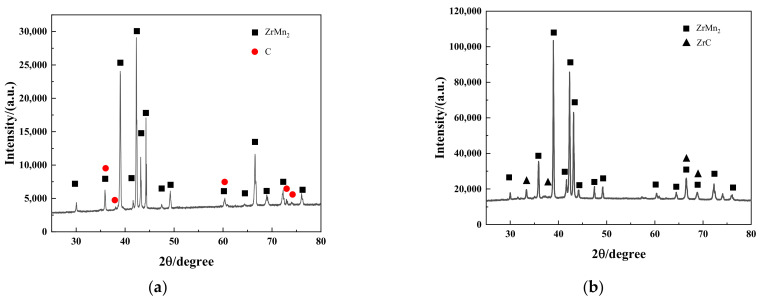
(**a**) XRD pattern of ZrMnFe alloy after purification performance test; (**b**) XRD pattern of ZrMnFe_0.7_Co_0.3_ alloy after purification performance test.

**Figure 9 molecules-28-04373-f009:**
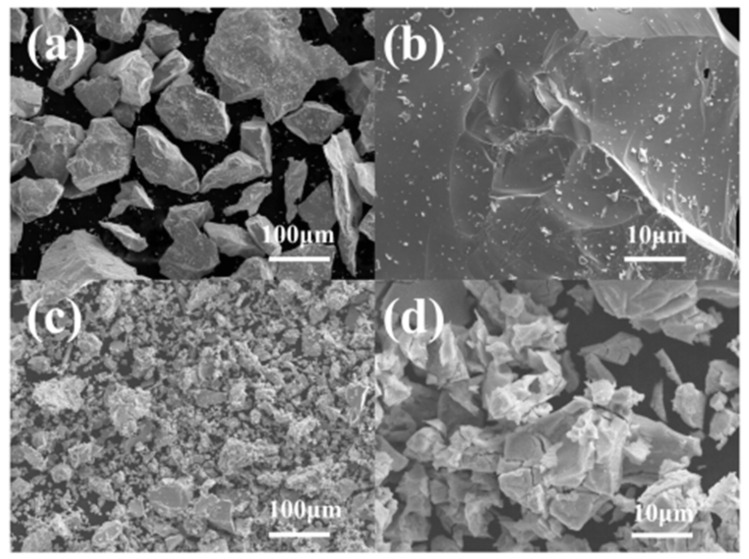
SEM images: (**a**,**b**) ZrMnFe alloy after testing; (**c**,**d**) ZrMnFe_0.7_Co_0.3_ alloy after testing.

**Figure 10 molecules-28-04373-f010:**
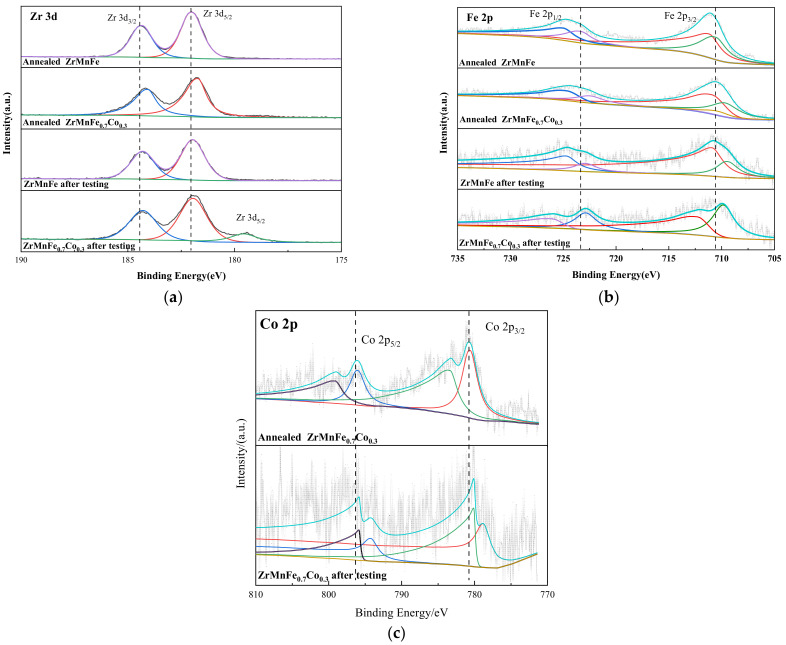
XPS spectrum: (**a**) Zr 3d; (**b**) Fe 2p; (**c**) Co 2p.

**Table 1 molecules-28-04373-t001:** Element content table of the ZrMnFe_1-x_Co_x_ (x = 0, 0.1, 0.2, 0.3, 0.4) alloy.

Sample	Zr	Mn	Fe	Co	Zr	Mn	Fe	Co
	Theoretical Value /wt%				Actual Value /wt%			
ZrMnFe	45.16	27.20	27.64	/	444.19	28.37	27.44	/
ZrMnFe_0.9_Co_0.1_	45.09	27.16	24.84	2.91	46.35	25.12	25.64	2.89
ZrMnFe_0.8_Co_0.2_	45.02	27.11	22.05	5.82	43.81	27.89	22.25	6.05
ZrMnFe_0.7_Co_0.3_	44.95	27.07	19.26	8.71	44.15	27.38	19.94	8.53
ZrMnFe_0.6_Co_0.4_	44.88	27.03	16.49	11.60	43.55	27.34	16.99	12.12

## Data Availability

Data available on request due to restrictions, e.g., privacy or ethical.

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
