# Peer review of "Enhancing Methane Removal Efficiency of ZrMnFe Alloy by Partial Replacement of Fe with Co"

_molecules, 2023, doi:10.3390/molecules28114373_

Round 1
Reviewer 1 Report
The abstract provides a clear overview of the research conducted, but it lacks some important details that should be included in a scientific abstract.
1. It does not provide a clear statement of the research problem or the research question that the study aimed to address. It is important to clearly state the research problem/question in the abstract to give the reader an understanding of the purpose of the study.
2. The abstract lacks information on the methods used in the study. It is important to mention the methodology used in the study to provide an idea of how the research was conducted and to help readers assess the validity of the findings.
3. The abstract should provide a brief summary of the main results of the study, including any statistical findings. This helps readers to understand the key findings of the research and assess the significance of the study.
Introduction:
1. While the introduction provides a broad overview of the importance of high-purity hydrogen in various applications and the detrimental effects of impurities such as methane, it lacks clarity and organization. The paragraph jumps from one application to another without properly introducing each one. Additionally, some of the statements are not supported by references, which weakens the credibility of the introduction.
2. The paragraph that discusses the various purification technologies suffers from similar issues. The list of purification technologies is not well-organized, and the advantages of using getters are not clearly stated until the end of the paragraph. Moreover, the paragraph could benefit from a brief explanation of how getters work, which would provide context for the reader.
3. The last paragraph of the introduction suffers from the same issues as the previous paragraphs. The statements are not well-supported by references, and the paragraph lacks clarity and organization. It could benefit from a more straightforward explanation of the challenges of purifying hydrogen at high temperatures and why enhancing the reactivity of the alloy with methane at low temperature is crucial for deep purification.
Materials and Methods
1. It would be helpful to provide more information on the specific alloys that were synthesized, such as their composition and intended use.
2. It would be beneficial to include more information on the equipment used during the purification process and how it was operated.
3. It would be helpful to include more information on the specific parameters used during each characterization technique, such as the XRD scanning rate and SEM magnification.
4. It would be helpful to include more information on the specific conditions used during testing, such as the temperature and pressure of the reaction vessel, and the gas flow rate during testing.
Results and Discussion
1. While the XRD pattern and SEM images provide some evidence for the successful synthesis and characterization of the alloys, more detailed characterization techniques such as TEM or HR-SEM could provide a more thorough analysis of the microstructure and composition.
2. The authors should provide more detail on the experimental setup for measuring the methane concentration at the outlet of the purification reactors. Specifically, how was the methane concentration measured and what was the uncertainty in the measurements?
3. The authors need to provide more discussion on the potential mechanisms behind the observed effects of Co substitution on the methane removal efficiency of the ZrMnFe alloys. Without a more thorough explanation, the observed trends seem somewhat arbitrary and difficult to interpret.
4. The authors should also discuss the limitations of the study, such as the relatively low test pressure and flow rate used in the experiments. Additionally, the authors should provide more information on the long-term stability and durability of the purification reactors, as well as any potential degradation or fouling mechanisms that could affect their performance.
5. The writing style could benefit from some additional clarity and conciseness. The authors should aim to present their findings in a more straightforward and accessible manner, rather than relying on complex or technical language that may be difficult for readers to understand.
Conclusions:
1. While the synthesis method appears effective in producing the desired phase structure and elemental content, the authors could provide more details on the characterization techniques used to confirm these results.
2. It's unclear whether the observed improvement in hydrogen purification performance is significant enough to warrant practical applications. The authors should provide a comparison to existing purification methods or standards to contextualize their results.
3. While the reduction in particle size due to Co substitution may contribute to the enhanced purification efficiency, the authors should consider other potential factors such as changes in surface area, composition, or catalytic activity. Additionally, they should provide evidence to support their claim that the Co substitution specifically lowers the formation energy barrier of ZrC.
4. The authors should provide more information on the conditions under which the observed enhancement in purification efficiency occurs. For example, does the same enhancement occur at different temperatures, pressures, or concentrations of impurities?
5. While the potential of these modified alloys for further development in the field of hydrogen purification technology is promising, the authors should provide suggestions or ideas for how this potential could be realized in practice. What are the next steps in advancing this technology?
The technical English writing should be improved.
Author Response
Dear Reviewer,
Thank you for your careful review of our manuscript. We appreciate your insightful comments and suggestions, which have greatly improved the quality of our work. We have carefully considered all of your comments and made the necessary revisions accordingly.
Please find our point-by-point response to your comments below:
Abstract:
- We have revised the abstract to provide a clear statement of the research problem and the research question that the study aimed to address. The main problem at present is that the commonly used ZrMnFe getter reacts with methane at a temperature as high as 700°C, and the removal depth is not enough. However, high-temperature purification of hydrogen is unsafe and energy-intensive, so it is necessary to reduce the temperature of the alloy to remove methane.
- The methods used in the study have been included in the abstract and marked with a yellow background.
- The main results of the study have been added, including the removal performance of the alloy for 10ppm methane in hydrogen and 50ppm methane in helium, and the reason why the Co substitution affects the performance. The content of the modified part is marked with a yellow background color.
Introduction:
-
Thank you for your valuable comments. We agree with your suggestions and have made revisions to the introduction. We have reorganized the paragraph and provided more details about each application of high-purity hydrogen. Moreover, we have added references to support the statements made in the introduction, which should enhance the credibility of the paper. We hope that the revised introduction provides a clear and organized overview of the importance of high-purity hydrogen and the detrimental effects of impurities such as methane.
-
We have added to the manuscript an introduction and discussion of the various purification methods and an introduction to how the getter works, which have been highlighted in the manuscript with a yellow background.
- We have added to the manuscript the necessity of improving the purification efficiency of getters at low temperature and marked with a yellow background.
Materials and Methods:
-
Content has been added and marked with a yellow background。
-
Content has been added and marked with a yellow background。
-
Thank you for your valuable feedback. We appreciate your suggestion to include more specific information about the characterization techniques used in our study. We have revised the manuscript to provide more details on the specific parameters used in each technique and have highlighted them with a yellow background.
-
Thank you for your valuable feedback. We have revised the manuscript and highlighted them with a yellow background。
Results and Discussion:
- Thank you for your comments and suggestions regarding our manuscript. We appreciate your feedback and understand your concern about the level of characterization in our study. In previous studies on getter materials, researchers have used XRD, SEM, and ICP to characterize the phase structure and composition of the alloy, and achieved successful results. We have followed the same approach in our study and provided evidence for the successful synthesis and characterization of the alloys using XRD and SEM. In the discussion with the supervisor, we think that for alloy materials, XRD structure refinement can provide a detailed characterization of the alloy structure. I have already applied to the testing center for urgent testing. However, during the May Day holiday period, it is required for employees to adjust their working hours in order to take five days off and then make up for the missed work on the weekend of the following week. I haven't received the test data yet due to the leave in lieu. I apologize for any inconvenience this may cause. Nevertheless, we appreciate your suggestion and will consider incorporating TEM or HR-SEM analysis in future studies, as we continue to improve our research.
-
The determination of methane concentration by gas chromatography is described in detail in 2.3 and marked with a yellow background. DID detector utilizes high-energy ultraviolet radiation (400-500nm) to ionize impurity molecules in the sample, and then collects and amplifies their signals through a collec-tor electrode, thereby obtaining the spectral peak of the measured component. During the test, the hydrogen standard gas containing 10ppm methane was used to make the standard spectrum, and then the gas passed into the outlet of the reactor was integrat-ed to obtain the response value, so as to obtain the methane concentration. The ionization voltage, current signal, carrier gas purity, and gas flow rate all have an impact on the test results. Therefore, adjusting the instrument parameters and conducting more than three tests with the raw gas is necessary. After the chromatographic peaks are stabilized, a standard spectrum should be created. Then, ppm-level methane should be measured by the instrument.
-
The removal of 50ppm methane in helium by the alloy replaced by Co proves the improvement of the reactivity of the alloy with methane. The XRD results prove that at the same temperature, the alloy replaced by Co is more likely to form metal carbides, and the carbon on the surface will not react with hydrogen to form methane again. The shift of the characteristic peak of Co in the XPS results indicates that the surroundings are in an electron-rich state, which is conducive to the breaking of the C-H bond of methane. The finding of XPS has been confirmed in previous studies, doi: 10.3866/PKU.WHXB201810052.
-
Limitations of the study are discussed in 3.2 and marked with a yellow background. Generally, in the actual application process, the gas purity will be monitored, and the purifier will be replaced in time when it is lower than the expected value, so the cost of materials must also be taken into consideration. Compared with the more researched palladium membrane purifier, the alloy purifier has a lower cost under the premise of obtaining the same hydrogen purity, but the working temperature is higher. Generally speaking, the factor that affects the performance of the getter material itself is mainly the catalytic activity. In the application of methane removal, the main cause of failure is the surface area carbon that leads to the re-reaction with hydrogen to form methane.
-
We have adjusted the language and used clearer and clearer images, and revised the manuscript in revision mode.
Conclusions:
-
Regarding the specific structure and composition analysis, we agree that techniques such as TEM can provide more detailed information. However, due to time constraints, we were not able to conduct TEM analysis in this study. We appreciate your suggestion and will consider incorporating TEM or HR-SEM analysis in future studies, as we continue to improve our research.
-
It was my negligence not to compare with other purification methods. I have added a comparison with other purification methods in the introduction, which may be more appropriate. The application of modified alloys is prospected and marked with yellow background.
- The XPS results demonstrated that Co substitution altered the electronic states of the surrounding elements, which led to improved catalytic activity of the alloy and hence enhanced purification efficiency. The activation energy of the gas-solid reaction can be calculated, but the use of a reactor other than a micro-bed reactor may result in significant errors in the calculation. Therefore, based on the XRD results, it can be inferred that ZrC did not form after the purification of the ZrMnFe alloy at the same temperature, while ZrC formed after the purification of the ZrMnFeCo alloy.
-
Purification efficiency as a function of temperature, pressure, gas flow rate has been added in the conclusion and marked with a yellow background.
-
The application of modified alloys is prospected and marked with yellow background. It is possible to consider the combination of the absorption agent and palladium membrane purification technology to produce higher purity hydrogen gas, in order to meet the demand for 9N purity gas in the semiconductor industry.
Reviewer 2 Report
The topic of this manuscript is methane removal from hydrogen by ZrMnFeCo alloy. There are some interesting results. The present referee recommends accepting the paper for publication after revision.
1. The ICP results presented in Table 1 for the ZrMnFe0.9Co0.1 alloy should be normalized for better accuracy.
2. The figures presented in the article, specifically Figures 4, 6, and 7, could benefit from a change in drawing style to enhance their visual appeal and clarity.
3. The authors should clarify the experimental conditions leading to the best results obtained after hydrogen purification. The current information is inconsistent, and further clarification is necessary.
4. The authors should revise the gas chromatogram to improve the clarity of the image to better convey the experimental results.
Author Response
We would like to express our sincere gratitude for your careful review of our manuscript titled Enhancing Methane Removal Efficiency of ZrMnFe Alloy by Partial Replacement of Fe with Co. Your insightful comments and constructive feedback have been immensely helpful in improving the quality of our work.
Based on your suggestions, we have made the following revisions to the manuscript:
1.ICP result for ZrMnFe0.7Co0.3 alloy has been normalized.
2.The plotting style of Figures 4, 6, and 7 has changed.
3.The best purification effect of ZrMnFe0.7Co0.3 alloy for hydrogen containing 10ppm methane is obtained at 500°C, 0.5MPa, 70sccm, and the outlet methane concentration is 0.215ppm.
4.The peak in the gas chromatogram should correspond to 0.292ppm CH4 in the hydrogen, which is the lowest outlet methane concentration corresponding to the temperature optimization. Gas chromatogram clarity has been improved.
Reviewer 3 Report
The manuscript investigates the effect of partial substitution of Fe with Co in ZrMnFe alloy for hydrogen purification, specifically for the removal of methane impurities. The manuscript presents an interesting approach to enhance the efficiency of hydrogen purification by partial substitution of Fe with Co in the ZrMnFe alloy. To further improve the quality of the paper, the authors should consider addressing the follwoing mentioned points:
1. Are there any characteristic peaks belonging to the Co species not observed in the XRD pattern? Please explain in which crystal phase Co exists.
2. Figure 5 is too blurry, please replace it with a clearer picture.
3. Please test the Raman spectra of the two samples ZrMnFe and ZrMnFe0.7Co0.3, and explain thewhereabouts of C in conjunction with Figure 8.
4. Please do BET test on two samples of ZrMnFe and ZrMnFe0.7Co0.3, according to the results to confirm the "The cracks caused the exposure of more active surfaces, thereby increasing the contact area between the reaction gas and the alloy surface" mentioned in line 293 ".
Some words need to be revised.
1. "performance" on line 47;
2. "temperatures" on line 90;
3. "were" on line 135 should be changed to "was";
4. "a" on line 144 should be changed to "an";
5. "effect" on line 244 should be changed to "effectiveness";
Author Response
Dear Reviewer,
We have revised the manuscript based on your valuable feedback. We are grateful for the time and effort you dedicated to the review process, and we believe that your suggestions have significantly improved the quality of our work.
According to your question, we make the following reply:
- The scanning speed used in the XRD test was 5°/min. The test results were then matched with the Jade software, and the matching degree of the characteristic peaks in the search and the ZrMn2 phase was found to be very high. Co partially replaces Fe in the C14 Laves phase. This phase is a close-packed cubic intermetallic compound with a MgZn2 structure, and it has been identified as the main phase in our study. However, we did not observe any characteristic peaks belonging to the Co species that were not detected in the XRD pattern.
- Figure 5 has been replaced with a higher resolution picture.
- I am aware that the Raman spectra data is critical for the manuscript, and I have already applied to the testing center for urgent testing. However, Raman spectroscopy needs to send samples to Beijing Normal University for testing, which takes a long time. The manuscript modification time is ten days, and I have not received the data due to the May Day holiday until the deadline. I apologize for any inconvenience this may cause. But the whereabouts of carbon can be explained by combining the book "Getters" and the research on St909 getter by Klein et al., it can be seen that the alloy undergoes surface catalytic cracking with methane, and due to the influence of concentration gradients, the surface carbon diffuses into the alloy at high temperatures to form metal carbides. In this study, the focus is on the improvement of the catalytic activity of the alloy methane reaction by substituting Co for Fe.
- Since the alloy is prepared by suspension induction melting method, the crystallinity is good, and the specific surface area is still very small even after crushing. The specific surface area measured by the ZrMnFe alloy is only 0.02m2/g, and the specific surface area measured by the ZrMnFe7Co0.3 alloy is only 0.12m2/g. This result cannot be ruled out due to instrumental testing errors.
Regarding the language issue you raised, it has been revised and marked in red font.
We appreciate your continued interest in our work and look forward to your feedback on the revised manuscript. Please let us know if you have any further comments or concerns.
Thank you again for your time and effort.
Round 2
Reviewer 1 Report
The revision has only had some improvement, but the quality of the paper is still not acceptable for publication.
Please proofread the entire manuscript.
Reviewer 3 Report
I appreciate for the efforts of the authors to revise their manuscript. It has been improved by incorporating the reviewers' comments and suggestions. It can be published as it is.